# Carcinoid Syndrome and Hyperinsulinemic Hypoglycemia Associated with Neuroendocrine Neoplasms: A Critical Review on Clinical and Pharmacological Management

**DOI:** 10.3390/ph14060539

**Published:** 2021-06-04

**Authors:** Francesca Spada, Roberta E. Rossi, Elda Kara, Alice Laffi, Sara Massironi, Manila Rubino, Franco Grimaldi, Sherrie Bhoori, Nicola Fazio

**Affiliations:** 1Division of Gastrointestinal Medical Oncology and Neuroendocrine Tumors, European Institute of Oncology (IEO) IRCCS, via G. Ripamonti 435, 20141 Milano, Italy; francesca.spada@ieo.it (F.S.); alice.laffi@ieo.it (A.L.); manila.rubino@ieo.it (M.R.); 2Hepatology and Hepato-Pancreatic-Biliary Surgery and Liver Transplantation, Fondazione IRCCS, Istituto Nazionale Tumori (INT), via G. Venezian 1, 20133 Milano, Italy; robertaelisa.rossi@gmail.com (R.E.R.); sherrie.bhoori@istitutotumori.mi.it (S.B.); 3Department of Pathophysiology and Transplantation, Università degli Studi di Milano, via Festa del Perdono 7, 20122 Milano, Italy; 4Endocrinology and Metabolism Unit, University Hospital S. Maria della Misericordia, Piazzale Santa Maria della Misericordia, 15, 33100 Udine, Italy; eldakara8882@yahoo.it (E.K.); doctor@francogrimaldi.it (F.G.); 5Division of Gastroenterology, San Gerardo Hospital, Bicocca School of Medicine, University of Milano Bicocca, 20126 Milano, Italy; massironi-400231@asst-monza.it

**Keywords:** carcinoid syndrome, insulinoma, neuroendocrine tumors, NENs

## Abstract

The carcinoid syndrome (CS) and hyperinsulinemic hypoglycemia (HH) represent two of the most common clinical syndromes associated with neuroendocrine neoplasms (NENs). The former is mainly related to the serotonin secretion by a small bowel NEN, whereas the latter depends on an insulin hypersecretion by a pancreatic insulinoma. Both syndromes/conditions can affect prognosis and quality of life of patients with NENs. They are often diagnosed late when patients become strongly symptomatic. Therefore, their early detection and management are a critical step in the clinical management of NEN patients. A dedicated and experienced multidisciplinary team with appropriate therapeutic strategies is needed and should be encouraged to optimize clinical outcomes. This review aims to critically analyze clinical features, evidence and treatment options of CS and HH and therefore to improve their management.

## 1. Introduction

Neuroendocrine neoplasms are rare and heterogeneous tumors originate from the cells of the diffuse neuroendocrine system. They frequently arise from the gastro-entero-pancreatic (GEP) tract, representing approximately 2% of all malignant tumors of the GEP system [1]. From a clinical point of view these neoplasms can be defined as functioning (F-NENs) or non-functioning neoplasms (NF-NENs) based on the presence of a clinical syndrome related to the secretion of various hormones released by the tumor itself [2]. Functioning forms generally represent approximately 30% of pancreatic NENs and 3–13% of patients with gastrointestinal NENs. Carcinoid syndrome (CS) and hyperinsulinemic hypoglicemia (HH) represent two of the most common clinical syndromes associated with NENs. In detail, the frequency of CS in the US Surveillance, Epidemiology, and End Results (SEER) database (2000–2011) was reported to be 19% in patients >65 years, with an increasing incidence over the years mirroring the general increase in frequency of NENs [3]. On the other hand, insulinomas (INSs), that is one of the most important causes of HH, occur in 1–4 people per million in the general population and represent 1–2% of all pancreatic neoplasms. However, their diagnosis is not straightforward, and it is often made when the patients are already strongly symptomatic.

The CS, the most frequent of the two [2], is a multi-organ disease that is characterized by a constellation of symptoms related to hormonal secretion and long-term complications, such as mesenteric fibrosis and carcinoid heart disease (CHD) [4,5,6]. The prevalence of CS has been reported to be 20–25% [5,7,8] and it occurs mainly in association with gastrointestinal (GI) or lung-related NENs [9].

An insulinoma (INS) is a functioning NEN originating from the neuroendocrine islet cells or multipotent stem cells of the pancreas, that produces insulin independently of glucose level’s stimuli. Insulinomas are therefore responsible for the HH syndrome [10,11] and, differently from CS that is usually associated with malignant NENs, are associated with a benign neoplasm with a 97% 5-year overall survival (OS) rate [12]. Indeed, malignant INSs represent less than 10% [10,11] of all INSs with an estimated incidence of 1–3 per million/year [13,14]. The diagnosis can be challenging and is generally delayed up to 5 years in approximately 20% of patients [10] due to the fact that symptoms are often misrecognized. In fact, despite the pathognomonic clinical presentation known as the “Whipple’s triad” [10,15]; due to its possible insidious symptoms often confused with other diseases, these patients are mistaken quite often as “mentally ill”. Once again, awareness and a prompt diagnosis represent the rate-limiting step of the whole therapeutic strategy in patients with F-NENs. Therefore, management by an experienced and dedicated multi-specialistic team is mandatory to offer patients the best therapeutic approach.

This review aims to address an overview on CS and INS from a multidisciplinary point of view, focusing on clinical aspects and drugs which are currently available in clinical practice or that are under clinical investigation, in order to critically improve patients’ management in the real-life scenario.

## 2. Material and Methods

Bibliographical searches were performed in PubMed, using the following keywords: carcinoid syndrome, insulinoma, neuroendocrine tumors, neuroendocrine neoplasms, diagnosis, therapy, guidelines. We also checked the Guidelines of the European Neuroendocrine Tumor Society (ENETS), American Society of Clinical Oncology (ASCO), North American Neuroendocrine Tumor Society (NANETS), European Society of Medical Oncology (ESMO), and Italian Association of Clinical Endocrinologists (AME). We searched for all relevant articles focused on F-NENs and published over the last 15 years. The reference lists from the studies returned by the electronic search were manually searched to identify further relevant reports. The reference lists from all available review articles, primary studies, and proceedings of major meetings were also considered. Articles published as abstracts were included, whereas non-English language papers were excluded.

## 3 Results

### 3.1. Pathophisiology and Clinical Features

#### 3.1.1. Carcinoid Syndrome

Over more than 40 humoral substances including histamine, prostaglandins, and tachykinins [16,17] were identified as being potentially involved in the pathogenesis of CS and leading to the development of the main clinical manifestations of CS (as reported in Appendix A) when they reach the systemic circulation. Nevertheless, the actual role of the different potential mediators in causing different symptoms of CS is still far from being clearly understood. In detail, as the majority of these substances are metabolized in the liver, in presence of liver metastases their metabolism is bypassed, leading to the development of CS that is, as a matter of fact, generally related to intestinal NENs metastatic to the liver [2,3,18]. However, in up to 13% of cases [19], CS can develop in patients without hepatic metastases, particularly in patients with NENs with primary sites in the ovary, testis, lung/bronchi, pancreas or occasionally, with GI NENs with large retroperitoneal metastases [3].

Serotonin (5-HT) has been proved to be the main mediator and 5-hydroxyindole-acetic acid (5-HIAA), that is the urinary metabolite of 5-HT, is the most accurate biomarker for CS. The overall sensitivity and specificity of urinary 5-HIAA in the presence of the CS are up to 90% [20]. It is the main indicator responsible for diarrhea development, given its effects on gut motility and secretion [3,18].

Elevated levels of 5-HT and other substances (neurokinin A, activin A, substance P, atrial natriuretic peptide, N-terminal pro-brain natriuretic peptide, chromogranin A (CgA), and connective tissue growth factor (CTGF)) are also implicated in the pathogenesis of fibrotic complications in the mesentery as well as in cardiac valves [21,22,23,24,25], the former leading to possible intestinal obstruction, the latter to CHD. However, again, both the specific mechanism underlying the occurrence of CHD as well as the individual substance contribution in the pathogenesis of fibrosis remain to be elucidated [21,26]. It has been reported that the tumor microenvironment (TME), that consists of supportive matrix, stromal, endothelial, and inflammatory cells, plays a relevant role in the development of fibrotic complications as it is crucial for tumor growth, invasion, and metastases onset.

A better understanding of the specific pathogenic mechanisms leading to the development of the typical symptoms of CS and fibrosis development is warranted in order to improve the clinical management of CS and its long-term complications by possibly identifying new targets for the treatment.

Around 80% of patients with CS present a chronic secretory diarrhea due to specific hormones that influence the GI motility inducing more than three bowel movements per day [27]. In some cases, diarrhea may lead to a dehydration and electrolyte impairment that can cause palpitations [28,29]. Along with diarrhea, almost 40% of patients with CS complain of abdominal pain attributable to increasing bowel movements, or as mentioned above, to mesenteric fibrosis associated with the tumor primary site.

About 50–85% of patients show skin flushing, a transient cutaneous erythema mainly localized on the face, neck, ear, and upper trunk and that is usually self-limiting after seconds or minutes [30].

Diarrhea and flushing may be provoked by the ingestion of some tyramine-containing food (e.g., cheese, coffee, alcohol, chocolate, dried or exotic fruits) or highly emotional states (anger, shyness) [31].

Finally, 10–20% of patients with CS show wheezing/asthma linked to the hypersecretion of several other peptides different from serotonin [32,33].

The “carcinoid crisis” (CC) is the most serious and acute complication of CS that is due to a peak of tyramine release. It may occur during particular stressogenic procedures such as surgery, biopsy or embolization. In this condition, changes in blood pressure (more often hypotension and shock), stupor, flushing, diarrhea, bronchospasm, hyperthermia and cardiac arrhythmias can occur. The carcinoid crisis is an emergency and life-threatening condition that needs to be promptly managed as it may lead to death [34]. Patients with F-NENs may also report some uncommon pathological conditions, such as pellagra, characterized by hyperkeratosis and pigmented skin lesions due to a deficit of niacin (B3 vitamin). In fact, in the presence of CS, tryptophan is used for the production of 5-HT, leading to a depletion of this precursor for its conversion to niacin [35].

The most important long-term complication of NEN patients with CS is CHD; it involves more than 50% of patients [24,36] and is due to the development of fibrotic plaques in the right heart valves and cavities, mostly resulting in tricuspid valve regurgitation.

Less commonly, thanks to the metabolism of vasoactive substances by the lung, the left side of the heart may be involved in cases of patent foramen ovale, pulmonary carcinoid disease or high burden of disease.

The diagnosis of CHD is usually delayed because its clinical signs appear when the right-sided heart failure is advanced. In these cases, the clinical symptoms and signs, which are lower-extremity edema, ascites, early satiety, and a reduced exercise capacity, represent the consequences of a central venous pressure increase due to raised jugular venous pressure [37]. Therefore, all patients diagnosed with CHD should be discussed within a dedicated and multi-specialist team including expert cardiologists and cardiac surgeons in an NEN-dedicated center.

#### 3.1.2. Insulinoma

Insulinomas are rare functional pancreatic neuroendocrine neoplasms (PanNENs) [13,38,39], but they are also the most common F-PanNENs [15], first described by RM Wilder et al. in 1927 [40]. Approximately 10% of INSs fall within the description of the multiple endocrine neoplasia type-1 syndrome (MEN1) [41]. Four characteristics of INS are all associated with a “90%” rate of presentation: 90% are solitary, 90% are benign, 90% occur in the pancreas, and 90% are less than 2 cm in diameter [42].

Insulinomas are the most common cause of endogenous HH in adults [43,44,45] and their differential diagnosis should also include pancreatogenous HH due to beta-cell hyperplasia (4% of all cases of HH) [46]—the condition being identified in patients who have undergone upper digestive bariatric surgery and in patients with idiopathic nesidioblastosis [47] and paraneoplastic endogenous hypoglycemia—and autoimmune hypoglycemia (insulin autoimmune syndrome or Hirata syndrome and type B insulin resistance) [47,48].

More than 99% of INSs are localized in the pancreas, but some cases are reported also in small intestine, pulmonary and cervix lesions [49]. Tumor cells produce insulin independently of glucose levels, consequently the clinical picture is characterized by symptoms of hypoglycemia, both autonomic and neuroglycopenic. Autonomic symptoms (i.e., palpitations, tachycardia, tremors and hunger) are correlated with the activation of an adrenergic and cholinergic response to hypoglycemia and appear when blood glucose is <60 mg/dL. Furthermore, when blood glucose is <45 mg/dL neuroglycopenic symptoms (that can vary from confusion to visual alterations, dizziness until seizures, and coma) can appear for very severe levels of hypoglycemia. Behavioral and personality changes, and weight gain due to continuous eating may occur [50,51], and these vague symptoms could be responsible for a diagnostic delay ranging from 1 month to many years. Moreover, over time we can observe an adaptation to low blood glucose levels, therefore, hypoglycemia-related symptoms will occur for very low glucose values [52,53]. Resolution of hypoglycemic symptoms after glucose supplementation is typical of INSs [41].

The symptoms of hypoglycemia are non-specific and for this reason it is important to confirm hypoglycemia by establishing the presence of the Whipple’s triad signs (hypoglycemia, plasma glucose levels <45 mg/dL, and resolution of symptoms after administration of glucose and correction of hypoglycemia) before starting a complex diagnostic work up for endogenous HH [15].

Among PanNEN patients, those with HH can be misunderstood or have an average delay between initial symptoms and final diagnosis of about 25 months [41]. However, HH can occasionally present during the follow-up of patients with originally metastatic NF-PanNEN, with a negative impact on the prognosis [54]. Malignant INSs, that exhibit local invasion and/or distal metastases, as a biological feature that differentiates malignant cases from benign ones [55], account for 10% of all cases of INSs [56]. Some pathological criteria could also be suggestive of malignancy, including lesions >2 cm, grading >G2 according to WHO 2019, or the presence of vascular invasion, perineural invasion or necrosis [57,58]. They are usually characterized by severe HH that is refractory to symptomatic treatment [56] and unlike benign INSs can cause prominent neuroglycopenic symptoms.

### 3.2. Diagnostic Algorithm

#### 3.2.1. Carcinoid Syndrome

The diagnosis of CS mostly derives from a clinical suspicion. Other causes of flushing (e.g., menopause related changes) and diarrhea (chronic inflammatory bowel diseases, travelers’ diarrhea, eating disorders) should be excluded first (Appendix A).

The diagnostic algorithm of CS includes initial screening for some biochemical markers and, if the clinical suspicion is confirmed, then the localization of tumors by the use of morphological/functional and, eventually, also endoscopic studies.

The 24-h urinary 5-HIAA level is elevated in up to 88% of cases of CS [59]. The sensibility and specificity of this marker are 100% and 85–90%, respectively; the international guidelines recommend strict dietary restrictions over three days before the assessment to avoid false positive results related to tryptophan-containing food [60,61]. The plasmatic 5-HIAA measurement seems not to be affected by food intake and allows CS diagnosis in a single assessment instead of the 24-h collection. However, it may be affected by a renal function impairment [62,63], thus, urinary 24-h 5-HIAA is still mostly used nowadays; however, considering that plasma fasting 5-HIAA measurement seems to correlate well with urinary levels and to achieve a similar diagnostic accuracy, in the near future, it will probably play a more relevant role [6].

Over the years, other substances have been explored as CS markers, including serotonin, which, however, is still not recommended as a standard diagnostic test due to the quick enzymatic conversion into 5-HIAA that makes the detection difficult [64].

Chromogranin A (CgA) is a glycopeptide that increases in >80% of NEN patients but, considering the risk of false-positive results related to drugs or clinical conditions, it is not indicated as an accurate diagnostic test for CS [65].

N-terminal pro B-type natriuretic peptide (NT-proBNP) represents a useful tool for the screening and follow-up of heart disease, being a good marker of right heart failure and cardiac dysfunction [66]. The use of NT-proBNP, together with an echocardiograph and cardiac magnetic resonance imaging (MRI) in some cases, is recommended for the diagnosis and follow-up of CHD [3,67].

In cases of positivity of the biomarkers, various morphological and functional imaging and, eventually, endoscopic procedures can be proposed for tumor localization.

Morphological imaging includes triple-phase computerized tomography (CT) scan and magnetic resonance imaging (MRI). Abdominal CT scan (with triphasic CT for the study of the liver) is the preferred diagnostic choice [68,69]. Some clinicians prefer MRI over CT because of its greater sensitivity for recognizing liver metastases [69].

Functional imaging can support clinicians in tumor characterization by the use of SRS scintigraphy or most recently 68Ga-DOTA-Peptide imaging, that has more than 80% sensitivity in the characterization of well differentiated NEN and has been repeatedly reported as more accurate than SRS scintigraphy in the detection of NENs [69,70].

Endoscopic studies of the upper or lower digestive tract with or without biopsy can be proposed in cases of positivity of the morphological or functional findings.

Carcinoid syndrome is often associated with small bowel NENs, as their diagnosis is known to be challenging, because of their non-specific presentation and the poor accessibility of the distal small bowel. In this specific setting the advent of video-capsule endoscopy (CE) and double-balloon enteroscopy (DBE) has improved the diagnosis of these neoplasms. However, nuclear medicine needs to be considered complementary to these endoscopic techniques and should be generally performed in the first instance, also considering that it might help to choose between the anterograde or the retrograde pathway as far as DBE is concerned [71].

#### 3.2.2. Insulinoma

Insulinoma should be suspected in a patient with autonomic and/or neuroglycopenic symptoms, blood glucose <55 mg/dL and insulin levels not suppressed in the presence of hypoglycemia.

The pathognomonic presentation is clinically defined by the Whipple’s triad [15] (Appendix A).

The diagnosis of hypoglycemia secondary to endogenous insulin secretion is biochemical and requires blood glucose <45 mg/dL associated with insulin ≥3 µUI/mL, c-peptide ≥0.6 ng/mL and proinsulin levels ≥5 pmol/L. As insulin secretion suppresses glycogenolysis despite hypoglycemia, ketones levels are normal and ß-hydroxybutyrate is ≤2.7 mmol/L. In severe hypoglycemic syndrome these criteria could be evaluated on a simple fasting blood test. When it is not possible, or when the clinical and biochemical parameters are not clear, the gold standard for the diagnosis is the 72-h fasting test [20,49]. Normally this test requires patient hospitalization and consists in keeping patients fasting for a maximum of 72 h while measuring glycemia, insulin, C-peptide, proinsulin and ß-hydroxybutyrate levels every 6 h, and every 2 h if finger stick or plasma glucose is <60mg/dL. The test ends when plasma glucose levels are <45 mg/dL with signs or symptoms of hypoglycemia, or plasma glucose <55 mg/dL even without signs or symptoms of hypoglycemia; only if the Whipple’s triad was previously documented or after 72 h of fasting. It is possible, at the end of the test, to perform a glucagon 1 mg infusion and if glucose increases to >25 mg/dL, this confirms the diagnosis of hypoglycemia secondary to endogen insulin secretion, as high insulin levels block glycogenolysis. Ninety-three percent of INSs develop symptoms in 48 h and 99% in 72 h. There are some rare cases of INSs that do not develop hypoglycemia during the test, and it is, sometimes, necessary to repeat the fasting test during the clinical follow-up.

To exclude a hypoglycemia secondary to administration of exogen insulin or sulfonylurea and glinides (factitious hypoglycemia), C-peptide and metabolites of oral antidiabetic drugs should be evaluated. Other common causes of hypoglycemia are reactive hypoglycemia that is typically post-prandial, and for overweight patients, post bariatric hypoglycemia consequent to gastric by-pass or gastrectomy that is also mainly post-prandial. Adrenal insufficiency is also a possible cause of hypoglycemia; in this case, hypoglycemia is associated with other symptoms, such as hypotension, nausea, abdominal pain, and intense asthenia and evaluation of adrenal function is necessary for differential diagnosis. Abdominal ultrasound is not routinely used because of its low sensitivity in lesion localization ranging from 0% to 86.5%. Better results can be achieved with contrast-enhanced ultrasound (sensitivity range 75–86.5%) [72]. Computed tomography (CT) with contrast and abdominal magnetic resonance (MRI) are often the initial studies of choice due to their good sensitivity for intrapancreatic lesions and lack of invasiveness. Nuclear imaging is widely used in NENs for tumor localization, characterization and for therapeutic purposes. Benign and malign INSs show a different pattern of SSTRs expression. Benign INSs have higher expression of SSTR 1.5 and 3 compared to SSTR 2 [73,74], so in this context sensitivity of SRS is relatively poor (<50%) [75,76]. On the other hand, benign INSs show a high expression of GLP-1R [77,78]. A widely use analogue of GLP-1, exenindine-4, has been shown to be superior to MRI and CT in the detection of INS with a sensitivity of 95% compared to 47% for CT/MRI [50,77,79]. On the contrary, malignant INSs often lack expression of GLP-1R but have high expression of SSTR, in particular SSTR 2 and 5 [46,50,75,80]. For this reason, Ga-DOTA-peptide imaging is currently used in malignant INSs for diagnostic and therapeutic purposes with a sensitivity of 87% [81].

Endoscopic ultrasound (EUS) plays a relevant role as a diagnostic tool to localize small (i.e., <2 cm) pancreatic lesions with a sensitivity and specificity of 75–100% and 95%, respectively for pancreatic tumors. The diagnosis of NEN can be confirmed by a fine needle aspiration/biopsy (FNA/B), even if false-negative results are possible mainly due to poor sampling adequacy. EUS-FNA/B is considered the primary sampling technique for pancreatic tumors, with a sensitivity ranging between 80% and 90%, specificity at 96%, and a sampling adequacy rate of 83–93% [82]. Methods with higher sensitivity are angiography with intraarterial calcium stimulation and venous sampling (ASVS), with a sensitivity of 85%. However, shortcomings of these methods are their invasive nature, with the concomitant risk of complications, and that they are less used, and only in high volume NEN centers [72,83].

Nuclear imaging is widely used in NENs for tumor localization, characterization and therapeutic purposes. The glucagon-like peptide 1 receptor (GLP-1R) is mainly expressed on the pancreatic beta cells and is therefore an interesting target for imaging of occult insulinomas. In various studies, the GLP-1 receptor agonists 111In-DOTA-exendin-4 and/or 68Ga-DOTA-exendin-4 PET/CT successfully detected benign insulinomas. GLP-1 receptor imaging has been reported to be superior to CT, MRI, and SPECT for the detection of small insulinomas [79]. However, as opposed to benign insulinomas, malignant insulinomas often lack the GLP-1R. Conversely, malignant insulinomas often do express the somatostatin receptor subtype 2 (SST2), which can be targeted using PET/CT using 68Ga-DOTA-labeled somatostatin analogs (SSAs), or with somatostatin receptor scintigraphy and SPECT. For that reason, Ga-DOTA-peptide imaging is currently used in malignant INSs for diagnostic and therapeutic purposes with a sensitivity of 87% [80]. This observation is clinically relevant since in cases of positive SSTR2 imaging, peptide radionuclide receptor therapy (PRRT) using 177Lu-DOTATOC or 177Lu-DOTATATE (Lutathera^®^) can be performed [79,84].

Finally, there are some retrospective studies also showing that (18)F-fluoro-dihydroxy-phenyl-alanine [18F] FDOPA positron emission tomography/computed tomography (PET/CT), combined with carbidopa premedication, could also be a valuable diagnostic tool in patients with occult insulinoma [85]. However, all these studies included small populations, thus further studies are necessary to clarify if [18F] FDOPA PET/CT could have a role in insulinoma detection [86].

### 3.3. Medical Management

#### 3.3.1. Carcinoid Syndrome

Treatment of CS is based on controlling specific signs and symptoms due to hormones released by the tumor itself, as well as treating the neoplasm.

##### Somatostatin Analogs

A cornerstone of CS treatment is the administration of SSAs, including octreotide and lanreotide, that inhibit the secretion of a broad range of hormones responsible for clinical symptoms.

They are highly effective at inhibiting the release of serotonin and other substances and diarrhea and flushing was significantly improved in over 80% of patients with the CS [87,88,89,90].

Similar efficacy on symptom control was observed with octreotide-long-acting repeatable (LAR) and lanreotide depot, with symptom reduction greater than 50% and with an overall CS control rate of 45.3% in patients with CS [87,91]. Somatostatin and its analogs are known to be able to inhibit different cellular functions, such as secretion, motility, and proliferation [92,93,94].

Two SSAs are commonly used in clinical practice: octreotide-LAR (long-acting-repeatable, Novartis, Basel, Switzerland) and lanreotide Autogel (Ipsen, Paris, France), which are administered by the intramuscular (i.m.) route and deep subcutaneous (s.c.) injections, respectively.

Given the pharmacokinetic profile of octreotide-LAR, the therapeutic dose is reached within 14 days. An induction scheme has been reported by some guidelines [95], although this regimen lacks evidence for a significant clinical advantage. On the other hand, lanreotide exhibits a different pharmacokinetic profile: it rapidly reaches therapeutic circulating levels in a few days, and then gradually decreases. No induction scheme is therefore needed and short-acting preparations for lanreotide are not available. An induction phase might be appropriate for octreotide only in cases of severe symptoms requiring immediate action, such as CC. In this case, even if the evidence for a real advantage is very low, the use of immediate-release octreotide as a ‘rescue’ medication has been suggested, and a reasonable starting dose might be 150 μg s.c. three times a day, even if some investigators prefer a continuous s.c. infusion of octreotide by a pump at a dose of 1000–2000 μg a day. The dose of immediate-release octreotide may be escalated when the maximum control of symptoms is achieved.

In general, focusing on symptom control, the optimal long-acting SSA dose is the smallest efficacious one. The licensed dosage of octreotide-LAR is 10, 20, or 30 mg every 4 weeks, and for lanreotide Autogel the recommended doses are 60, 90, or 120 mg every 4 weeks. However, the only dose of octreotide that has shown an anti-proliferative effect is 30 mg, from the PROMID study [96,97] and the only dose of lanreotide Autogel is 120 mg, from the CLARINET study [98].

Somatostatin analogs exhibit a good safety profile, although some adverse effects might occur, including abdominal pain with cramps, constipation, diarrhea, steatorrhea, injection site irritation, local pain, nausea, and vomiting. Biliary stone disease is reported as a common side effect of SSAs (i.e., 27% with almost 28% of the patients developing biliary complications) as recently reported by a multicentric study [99]. More rarely hypothyroidism, acute pancreatitis, alopecia, acute hepatitis, hyperbilirubinemia, hyperglycemia, hypoglycemia, and prolonged QT interval are also described as possible complications.

##### Non-Conventional Doses of Somatostatin Analogs (SSAs)

Patients with persistent or recurrent flushing and/or diarroea may benefit from non-conventional doses of SSAs, with higher doses or more frequent administration of long-acting SSAs. In a systematic review, eleven out of the seventeen included studies reported the use of higher doses of SSA (>30 mg/monthly) to control symptoms and tumor progression; of these, ten studies reported efficacy with doses ranging from 40 mg/monthly or 30 mg every 3 weeks up to 120 mg/monthly, with no significant toxicity [100]. These results were confirmed by a more recent and high-quality systematic review, based on eighteen studies (with 1002 patients), which reported significant rates (30–100%) of disease control with high-dose SSAs, but low rates (0–14%) of tumor response, with symptom improvement in a variable percentage (namely from 23 to 100%) [101]. Furthermore, according to a recent study investigating the real-world use of SSAs in NEN patients among the Italian medical oncological community, more than 70% had used a shorter interval between administrations, up to 3 or even 2 weeks in functioning forms [102].

According to most guidelines and experts’ opinions, this treatment option should be considered for patients with refractory CS [103].

##### Telotristat Ethyl (TE)

For refractory symptoms, telotristat ethyl (TE) can be used in combination with SSAs to control diarrhea associated with CS. The TELESTAR trial was a three-arm study evaluating two doses of oral TE (250 mg and 500 mg, each taken three times daily) against placebo, conducted in 135 patients with a history of CS who were receiving treatment with SSAs but had uncontrolled diarrhea (≥ 4 bowel movements daily). TE at either dose was associated with a statistically significant reduction in the frequency of bowel movements compared with placebo [104]. Also, the TELECAST phase III trial supported the safety and efficacy of TE when added to SSAs in patients with CS-related diarrhea [105]. Finally, the effectiveness of TE was also evaluated in the real-world setting using patient-reported data from a nurse support program over 3 months, confirming a significant improvement mainly in the bowel movement frequency, with the addition of TE treatment [106]. Even if TE could improve other symptoms such as abdominal pain, nausea, and flushing, it remains unclear whether TE has a significant impact on flushing, as this symptom is not directly related to serotonin.

##### Interferon Alpha

Interferon-alpha (IFNα) is a cytokine that plays an immunomodulatory, antiproliferative, and antiviral role. It has been used for years alone or in combination with chemotherapy and SSAs to treat metastatic NENs [107]. A relatively recent retrospective study confirmed its importance and effectiveness in 35 patients treated with IFNα with a median PFS of 25 months, 3% complete response, 3% partial response, and 71% stable disease [108]. Due to the advent of new therapies, it has largely fallen out of favor, even if there remain some patients that may benefit from IFNα as either an adjunct to treatment or a bridging therapy while awaiting commencement of another treatment. IFNα is given as a subcutaneous injection, most often at a dose of 3–5 million units (mU) three times a week or alternatively as weekly injections of 75–150 μg long-acting pegylated (PEG)-IFNα. Side-effects include initial flu-like symptoms, chronic fatigue, depression, anemia, and neutropenia. Besides, autoimmune responses are manifested in 15–20% of patients, most commonly thyroid dysfunction.

##### Other Antidiarrheal Therapy

Antidiarrheal agents, such as loperamide, can be used for control of refractory diarrhea. A few reports demonstrated that serotonin receptor antagonists, such as ondansetron, can alleviate diarrhea in patients with refractory CS [109,110,111]. Moreover, in the setting of a refractory syndrome, other possible causes of diarrhea should be considered [112]. It is, indeed, well known that SSA therapy itself can cause fat malabsorption and steatorrhea that may respond to pancreatic enzyme supplementation. Again, patients who have undergone resection of the distal small bowel frequently develop bile malabsorption and bile salt-induced diarrhea that can be treated with cholestyramine.

#### 3.3.2. Insulinoma

The prevention and treatment of hypoglycemia is based on a rigorous education of the patient and patient’s family/friends in symptom recognition and treatment. Lifestyle modifications should include frequent meals rich in complex carbohydrates throughout the day and evening, and avoiding excessive exercise and driving [113].

It is particularly important to prevent prolonged periods of fasting by consuming small, frequent meals during the day and night. Moreover, as the degradation is slower, assumption of complex carbohydrates can help in maintaining stable blood glucose. When hypoglycemia is very severe, parenteral or enteral nutrition may become necessary [114].

Pharmacological management of hypoglycemia depends on the patients’ and tumor’s characteristics.

##### Diazoxide

Diazoxide is an antihypertensive benzothiadiazine derivate used as a first-line treatment for glycemic control in patients with INS preoperatively, or in cases of persistent hypoglycemia after surgery [38,115,116]. It inhibits insulin secretion in pancreatic beta cells [115,117], increases hepatic gluconeogenesis and inhibits skeletal muscle glucose uptake [118]. The therapeutic initial daily dosage in adult patients is 50–300 mg divided into two or three administrations [119]. Then, depending on the achieved glucose levels, the dosage can be increased up to a maximum of 600–800 mg per day, especially in refractory hypoglycemia due to malignant INSs [10]. A clinically relevant therapeutic effect is usually achieved in several days, but if this doesn’t occur in 2–3 weeks the treatment should be discontinued [119]. Possible adverse effects during treatment include edema due to fluid retention (which is the most common side effect and can be treated with thiazide diuretics), hirsutism, weight gain, diarrhea, nausea, vomiting, rash, headache, abdominal pain, palpitations, and kidney malfunction [120,121,122]. In order to prevent and control nausea, diazoxide should be taken with a meal [10]. Adverse events are usually not severe and do not require interruption or modification of the dosage, as the therapeutic benefit exceeds the side effects [120,121]. Long-term treatment with diazoxide seems to be safe [121,123]. Diazoxide is contraindicated during pregnancy since it is teratogenic in animals, whereas there are no data about its transmission to breast milk [116]. Simultaneous treatment with SSAs is not recommended since SSAs seem to inhibit diazoxide hyperglycemic effects [115].

##### Long-Acting Somatostatin Analogs (SSAs)

Long-acting SSAs are considered a second-line treatment for hypoglycemia in patients with INS and they are especially used in patients who cannot undergo surgery and when diazoxide is not recommended due to adverse effects or inefficacy [113]. For their well-known anti-proliferative effect, SSAs can also be considered first-line treatment in malignant INSs [42,115]. The hyperglycemic effect of SSAs is mediated especially by SSTR-2 receptors [115]. Accordingly, this effect cannot occur in INSs that do not express SSTR-2 receptors. If SSTR-2, SSTR-3 and SSTR-5 are not expressed or have a low tumor expression, SSAs can lead to paradoxical hypoglycemia because they also inhibit counterregulatory hormone secretion (i.e., glucagon and growth hormone) [115,117]. In order to prevent this paradoxical effect, the treatment should be initiated with a short-acting subcutaneous form of SSA during hospital observation [42,115,117]. Then, after switching to a long-acting form, the short-acting one can be used in cases of exacerbation of hypoglycemia [95].

In cases of hypoglycemia refractory to SSAs, the compassionate use of pasireotide, a second generation, multireceptor or so-called pan-receptor SSA, may be considered even though it is not registered for use in PanNENs [124]. It has high affinity for SSTR-1, SSTR-2, SSTR-3 and most importantly SSTR-5 and presents hyperglycemia as a common adverse effect, but experience in malignant INSs is very limited [125,126,127].

##### Everolimus

Everolimus can also be used for refractory hypoglycemia [49,56]. One of the most common adverse effects observed with everolimus is hyperglycemia and even though in the randomized clinical trials of JC Yao et al. there were only a few cases of INSs, a retrospective analysis of these INSs [128] showed an improvement of glycemic control during treatment with everolimus. The efficacy in glycemic control is only partially due to its effect on tumor mass [113]. The mechanisms that can sustain hyperglycemia induced by everolimus include the reduction of production and secretion of insulin, the increase of peripheral insulin resistance, and a possible increase of hepatic gluconeogenesis [129,130]. Everolimus is administered at a dose of 10 mg per day and the clinical effect is usually observed within two weeks [131,132,133,134,135].

##### Glucocorticoids

Glucocorticoids can induce hyperglycemia as an adverse effect; therefore, they were used for glycemic control before better pharmacological alternatives were available. Among them, glucocorticoids and especially prednisone, help glycemic control by inhibiting insulin synthesis, increasing insulin resistance, and stimulating gluconeogenesis. However, glucocorticoids have important adverse effects, especially in the long term, such as hypertension, osteoporosis and vertebrae fractures, edema, and increased susceptibility to infections [49].

##### Glucose

Glycemic control is particularly important during surgical procedures, ablative therapies and peptide receptor radionuclide therapy (PRRT) since all of these therapies can induce a high risk of severe and life-threatening hypoglycemia. For this reason, an intravenous infusion of glucose solution and glucose boluses along with SSAs (both short and long acting) can be used during and after the treatment. Despite the administration of glucose, oral or parenteral depending on the level of consciousness, in cases of recurrent refractory hypoglycemia, nocturnal enteral feeding should also be considered. In the event of a lack of intravenous access or acute severe hypoglycemia in unconscious patients, glucagon (intramuscular, subcutaneous injection or intranasal spray) can also be used in an emergency. The intranasal spray formulation is ready for use and it doesn’t require previous training for administration [136]. On the other hand, promising results have been observed with PRRT in controlling hypoglycemia [137].

Continuous glucose monitoring systems (CGMS) in patients with INSs can detect hypoglycemia (particularly relevant in patients with unawareness of hypoglycemia), monitor the achieved glycemic control with medical therapy, and confirm biochemical remission after surgery [42]. These techniques are considered important since they alert patients to low glucose concentrations before they present neuroglycopenic symptoms. These systems are connected to a computer-regulated infusion pump that, based on previously defined blood glucose values and depending on the level of glucose, automatically administer the right amount of insulin or glucose. In this context CGM devices are used especially for glucose administration, when hypoglycemia occurs [42,138,139].

Currently, there is not a defined therapeutic sequence for refractory hypoglycemia in malignant INSs. This is another reason that makes a multidisciplinary approach in referral centers of paramount importance for patients with malignant INSs.

#### 3.3.3. Other

For patients with rapidly progressive and/or symptomatic NENs, greater tumor control can be obtained with cytotoxic chemotherapy (agents such as 5-fluoruracil, streptozotocin, capecitabine, temozolomide, dacarbazine), liver directed therapies, PRRT and then, everolimus and sunitinib. However, their specific role in functioning tumors should be further investigated in a more specific and well-selected population with well-designed studies.

As for chemotherapy, temozolomide-based chemotherapy can be used in rapidly progressive or high tumor burden well differentiated NENs mainly from the pancreas. Although the low level of literature evidence due to the heterogeneity of the selected population, the type of studies, mainly retrospective, and the potential risk of biases, temozolomide-based chemotherapy seems to be active, including for functioning NEN patients [140].

Even in the absence of specific randomized trials among the various loco-regional therapies, liver directed therapy (LDT), and especially hepatic artery embolization, represent a good option in patients with functional NEN, based on the patient’s characteristics and the physician’s expertise [141].

Moreover, in patients with hormone-excess-state, independent of its anti-tumor effect, PRRT is particularly promising and based on a phase III randomized trial (NETTER-1) comparing ^177^Lu-DOTATATE to high-dose octreotide (60 mg/month), has already been approved for progressive functioning and not-functioning midgut NENs. Moreover, treatment with ^177^Lu-DOTATATE resulted in a safe and effective therapy with radiological, symptomatic and biochemical responses in a high percentage of patients with metastatic functioning PanNENs [137,142].

Everolimus plus octreotide long-acting repeatable (LAR) has been studied in patients with low-grade or intermediate-grade NENs originating from various primary sites and with a history of secretory symptoms (diarrhea or flushing) attributable to carcinoid syndrome [143]. Although the clinical results are interesting, the primary endpoint in terms of PFS assessed by central radiology review was not statistically significant; therefore, the study is considered negative. However, from a practical point of view, the favorable trend in improving PFS as per a local investigator review, should be further investigated by not excluding a positive clinical impact absolutely. Moreover, some experiences highlight the ability of everolimus to regulate glycemic levels in patients with INS [128,144].

Finally, sunitinib is the only tyrosine kinase inhibitor approved for the treatment of PanNENs based on a phase III trial published in 2011 [145]. Among the 171 patients enrolled, there were 85 functioning PanNENs with various kinds of syndromes. However, the authors found that there was not a significant difference in terms of progression free survival (PFS) among the group of patients in the sunitinib group versus placebo, and regarding various parameters including functional status. Therefore, currently we don’t know really if the favorable results of this study could be applied to functioning PanNENs given that some other experience seems to suggest a detrimental impact of sunitinib in the management of INSs [146].

## 4. Discussion

Paraneoplastic syndromes were described about 100 years ago and, over the time, many forms of endocrinological interest have been identified that can cause CS, Cushing’s syndrome, syndrome of inappropriate ADH secretion, hypoglycemia, hypercalcemia, and acromegaly, depending on the substances produced by the neoplasm.

These syndromes, typically associated to NENs and often diagnosed before the associated tumor, depend on the possibility and heterogeneous capability of these neoplasms to produce, secrete, and release biologically active substances that can cause several clinical conditions. Furthermore, monitoring the various peptides can be a way to monitor the tumor response to therapy. In most cases the various substances are produced by the tumor in ectopic sites and seem to mimic those normally produced by our organs, therefore the physician’s awareness on diagnosis and management of these syndromes in patients with NENs is challenging. Considering the increase in the incidence of NENs, it is likely that the incidence of the paraneoplastic syndromes related to NENs will also soon be increasing, so identifying and registering such cases is crucial to develop evidence-based diagnostic and therapeutic guidelines for their management.

Carcinoid syndrome, usually secondary to a small intestinal NEN metastatic to the liver and HH caused by INS, are two of the most important NEN-related conditions that clinicians face in their everyday clinical practice. The diagnosis of these two syndromes is, however, particularly challenging due to their non-straightforward and misleading clinical presentation. In detail, patients with CS might, due to the presence of chronic diarrhea, be referred to a general gastroenterologist and mistakenly diagnosed with irritable bowel syndrome (IBS) or inflammatory bowel disease (IBD), whereas patients with INS, who tend to present with insidious symptoms, might be often labeled as “mentally ill”. In both cases the diagnosis is often delayed with a negative impact on the global NEN prognosis and on the quality of life of these patients.

Regarding treatment options, certain progress has been made over the years and new treatment strategies have become available with promising results in terms of symptomatic control and consequent improvement in patients’ quality of life. According to two recent phase III studies from Kulke et al. and Pavel et al. mentioned above, TE has been reported as a safe and effective treatment option when added to SSAs in patients with CS-related diarrhea, as well, encouraging results have been presented for non-conventional, high-dose SSAs. In the setting of malignant INSs, although a defined therapeutic sequence for refractory hypoglycemia is still not available, the compassionate use of pasireotide as well as treatment with everolimus, might be considered as viable options for hypoglycemia refractory to SSAs.

Even if many reports and guidelines regarding F-NEN management are available and new drugs for clinical management have been developed, the awareness of clinical syndromes seems to be a prerogative of single specialists such as gastroenterologists, endocrinologists or oncologists, based on specific clinical presentation; however, an integrated approach among these specialties to improve clinical management and outcomes, already working well in many hospitals, could be better implemented in some others.

On one hand, all over Europe there are many Centers of Excellence for GEP-NEN management certified by the ENETS, but on the other hand, there are also hospitals and university centers, in which there are mono- or oligo-specialists skilled on NENs.

Despite the availability of such expertise, the diagnostic/therapeutic pathway of a patient with F-NEN relies mostly on the vision of a single doctor, or on the choice of the patient and/or family to seek second opinions elsewhere. Furthermore, it is not uncommon that a patient with an F-NEN is managed in non-NEN specialized or dedicated structures.

However, a validated diagnostic/therapeutic path for patients with NEN, functioning or non-functioning, has not been yet defined, above all under bureaucratic/structural terms. From a clinical point of view, the benefit of the patient is strongly correlated to a multi-specialistic diagnostic/therapeutic path coordinated by a dedicated multidisciplinary team (MDT), that should step in as soon as possible in the patient’s clinical history.

## 5. Conclusions

Given the rarity and clinical and biological heterogeneity of NENs, experience, competence, and centralization of case studies can play an important role. It is conceivable that a team of specialists, all dedicated to NENs, who simultaneously discuss the patient with just a suspicion or with the first appearance of a related clinical syndrome, is the best diagnostic/therapeutic approach that should be provided. Future collaborative studies conducted within the NEN-referral centers network and specifically focused on CS and HH and their complications are warranted and desired.

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
