# Peer review of "Carcinoid Syndrome and Hyperinsulinemic Hypoglycemia Associated with Neuroendocrine Neoplasms: A Critical Review on Clinical and Pharmacological Management"

_pharmaceuticals, 2021, doi:10.3390/ph14060539_

Round 1

Reviewer 1 Report

I completely agree with the conclusion phrase the authors write “It is conceivable that a team of specialists, all dedicated to NENs, who simultaneously discuss the patient with just a suspicion or with at the first appearance of a related clinical syndromes or their suspicion, is the best diagnostic/therapeutic approach that should be provided”.

In the same way also pubblications need to have the same presence of specialised authors. A highly professional competence has to be present to perform reviews, draw algorithms and write papers expecially if the publication has an educational intent. The diagnostic section, too short and inadequate, has to be completely added.

A large chapter of possible therapies, has completely ignored PRRT, Captem, temozolomide. If exaustive publications are lacking, difficult to believe, at least an experienced and competent impression has to be offered to the readers.

I would also add an explanation of a possible paraneoplastic origin of the carcinoid syndrome or hyperinsulinemic hypoglicemia. This frequent possibility has to be clarified not only in the discussion section. In the same way, explain why the carcinoid hearth disease doesn’t interest the left hearth portion.

 Line 51, 58,64 to be better clarified

Explain that there is an adaptation to hypoglycemia in syndromic patients who are clinically less affected by low blood glucose values. Line 168 paragraph

line 214 clinically

line 275 repeated sentence

Author Response

I completely agree with the conclusion phrase the authors write “It is conceivable that a team of specialists, all dedicated to NENs, who simultaneously discuss the patient with just a suspicion or with at the first appearance of a related clinical syndromes or their suspicion, is the best diagnostic/therapeutic approach that should be provided”.

Point 1: In the same way also, publications need to have the same presence of specialized authors. A highly professional competence has to be present to perform reviews, draw algorithms and write papers especially if the publication has an educational intent. The diagnostic section, too short and inadequate, has to be completely added.

Response 1: Thank the reviewer for pointing out this deficiency. We have added a paragraph on diagnosis in the section 2.2.1 on CS and another one paragraph in the section 2.2.2 on INS. Please see rows 189-191 and rows 213-223. We also revised the Figure S1, please see the row 554.

Point 2: A large chapter of possible therapies, has completely ignored PRRT, Captem, temozolomide. If exhaustive publications are lacking, difficult to believe, at least an experienced and competent impression has to be offered to the readers.

Response 2: Thank the reviewer for pointing out this deficiency. We have added a paragraph on Interferon alpha in the section 2.3.1, then on PRRT, CAP-TEM / TEM in the section 2.3.3. Please see the rows 521-567.

Point 3: I would also add an explanation of a possible paraneoplastic origin of the carcinoid syndrome or hyperinsulinemic hypoglicemia. This frequent possibility has to be clarified not only in the discussion section. In the same way, explain why the carcinoid hearth disease doesn’t interest the left hearth portion.

Response 3: Thank the reviewer for pointing out this point. For the specific details:

  • For carcinoid syndrome we have specified that: “Less commonly, thanks to the metabolisms of the vasoactive substances by the lung, the left side of the heart may be involved in cases of patent ovale foramen, pulmonary carcinoid disease or high burden of disease” (rows 147-149) above the already present rows 82-93.
  • For insulinomas we have specified the “paraneoplastic endogenous hypoglycemia” as differential diagnosis with insulinoma (Please see the rows 168).

Point 4: Line 51, 58,64 to be better clarified

Response 4: Thank the reviewer for highlighted these points. Row 51 à 57-59: we have eliminated part of the sentence “…being often associated to liver metastases from small bowel NENs” because it was misleading, and the pathogenesis of carcinoid syndrome is better specified in the pathophysiology section 2.1.1. Please see the rows 63-70: we have rephrased the sentence.

Point 5: Explain that there is an adaptation to hypoglycemia in syndromic patients who are clinically less affected by low blood glucose values. Line 168 paragraph.

Response 5: Thank the reviewer for this comment. We added in line 180 “Moreover, overtime we can observe an adaptation to low blood glucose level therefore, hypoglycemia-related symptoms will occur for very low glucose values.” Please see the rows 180-182.

Point 6: line 214 clinically

Response 6: Thank the reviewer to have observed this grammatical mistake. We have revised accordingly.

Point 7: line 275 repeated sentence

Response 7: Thank the reviewer for this comment. We have eliminated the repeated sentence and revised accordingly.

Reviewer 2 Report

The aim of this review is to describe the clinical features of carcinoid syndrome and hyperinsulinemic hypoglicemic syndrome due to insulinoma “from a multidisciplinary point of view, focusing on clinical aspects and drugs which are currently available in clinical practice or that are under clinical investigation”. The manuscript is well organized and comprehensively described, and the topic is of interest. In my opinion there are some minor points that should be addressed.

Specific issues

  • In the diagnostic algorithm for insulinoma (paragraph), PET/CT with 68Ga-labeled peptides (68-Ga-DOTATOC/DOTATATE) is not mentioned at all, whereas in Figure S2 is only mentioned for metastatic insulinomas. I think that the Authors should dedicate some sentences to this option.
  • In the “Medical management” paragraph, PRRT is not included among the treatment options. I think that also this topic deserves to be mentioned. Furthermore, other drugs should also be mentioned to be exhaustive for a review on the treatment of pancreatic NENs (eg sunitinib, chemotherapy…)
  • Little attention is deserved to diet for the management of hypoglycemia (Lines 353-355), whereas it is not mentioned at all for CS symptoms. I suggest strengthening these parts (eg, “The management of neuroendocrine tumours: A nutritional viewpoint”. NIKE Group. Crit Rev Food Sci Nutr. 2019;59(7):1046-1057. doi: 10.1080/10408398.2017.1390729).
  • Paragraph “Everolimus”: This paragraph deals also with glucocorticoids, so I think that it should be spit in two or renamed. Furthermore, everolimus has been proposed as first line therapy for patients with symptomatic malignant unresectable insulin-secreting pancreatic NENs. This point could deserve a comment by the Authors.
  • Lines 413-414: “Moreover, nowadays the simultaneous use of glucocorticoids and everolimus is contraindicated”. Please, add a reference for this statement.
  • Lines 433-435: “These systems [CGMS] are connected to a computer-regulated infusion pump that, based on previously defined blood glucose value that is previously defined, depending on the need, automatically administer the right amount of insulin or glucose”. Not always true: CGM devices are often used without insulin pumps.
  • Lines 479-481: “an integrated approach among these specialties to improve clinical management and outcomes is still missing.” I think that this integrated approach could be better implemented in some hospitals, but it is already working well in many others.

Minor issues

  • Just from the abstract and throughout the manuscript, neuroendocrine tumors are abbreviated both as NETs and NENs. I suggest using just one acronym (eg NENs, more comprehensive and updated term)
  • Abstract, line 21: “The former is mainly related to the serotonin secretion by a small bowel NET, in most cases located in the small bowel” (redundancy of “small bowel”)
  • Acronyms should be explained the first time they are used, and therefore should be used along the text:
    1. Introduction, Line 47: Carcinoid syndrome (to be replaced by CS)
    2. Introduction, Line 57: insulinoma (INS) could already be explained at Line 44, and therefore INS should be used throughout the manuscript
  • Discussion, line 492: “From a clinical point of view, the clinical benefit of the patient is strongly correlated…” (redundancy of “clinical”).

I think that a recent review dedicated to CS deserves to be mentioned: Serotonin pathway in carcinoid syndrome: Clinical, diagnostic, prognostic and therapeutic implications. NIKE Group. Rev Endocr Metab Disord. 2020 Dec;21(4):599-612. doi: 10.1007/s11154-020-09547-8.

Author Response

The aim of this review is to describe the clinical features of carcinoid syndrome and hyperinsulinemic hypoglicemic syndrome due to insulinoma “from a multidisciplinary point of view, focusing on clinical aspects and drugs which are currently available in clinical practice or that are under clinical investigation”. The manuscript is well organized and comprehensively described, and the topic is of interest. In my opinion there are some minor points that should be addressed.

Specific issues:

Point 1: In the diagnostic algorithm for insulinoma (paragraph), PET/CT with 68Ga-labeled peptides (68-Ga-DOTATOC/DOTATATE) is not mentioned at all, whereas in Figure S2 is only mentioned for metastatic insulinomas. I think that the Authors should dedicate some sentences to this option.

Response 1: Thank the reviewer for this suggestion. We have added a paragraph on nuclear imaging technique in the diagnostic algorithm of insulinoma, paragraph 2.2.2, please see the rows 390-400.

Point 2: In the “Medical management” paragraph, PRRT is not included among the treatment options. I think that also this topic deserves to be mentioned. Furthermore, other drugs should also be mentioned to be exhaustive for a review on the treatment of pancreatic NENs (eg sunitinib, chemotherapy…)

Response 2: Thank the reviewer for pointing out this deficiency. We have added a paragraph 2.3.3 “Other” on temozolomide-based chemotherapy, liver directed therapies, PRRT, everolimus/sunitinib to explain the role of these therapies in functioning NETs. Please see the rows 592-634.

Point 3: Little attention is deserved to diet for the management of hypoglycemia (Lines 353-355), whereas it is not mentioned at all for CS symptoms. I suggest strengthening these parts (eg, “The management of neuroendocrine tumours: A nutritional viewpoint”. NIKE Group. Crit Rev Food Sci Nutr. 2019;59(7):1046-1057. doi: 10.1080/10408398.2017.1390729).

Response 3: Thank the reviewer for this suggestion, we added the following sentence: “Particularly is important to prevent prolonged periods of fasting by consuming small, frequent meals during the day and night. Moreover, as the degradation is slower, assumption of complex carbohydrates can help maintaining stable blood glucose. When hypoglycemia is very severe parenteral or enteral nutrition may become necessary. “

Point 4: Paragraph “Everolimus”: This paragraph deals also with glucocorticoids, so I think that it should be spit in two or renamed. Furthermore, everolimus has been proposed as first line therapy for patients with symptomatic malignant unresectable insulin-secreting pancreatic NENs. This point could deserve a comment by the Authors.

Response 4: Thank the reviewer for this suggestion, we have revised accordingly. Please, see the paragraphs splitted at rows 463-482.

Point 5: Lines 413-414: “Moreover, nowadays the simultaneous use of glucocorticoids and everolimus is contraindicated”. Please, add a reference for this statement.

Response 5: Thank the reviewer for this suggestion, we have deleted that sentence, as in clinical practice everolimus in used with glucocorticoid, especially in patients with metastatic insulinoma. Please, see the rows 482-483.

Point 6: Lines 433-435: “These systems [CGMS] are connected to a computer-regulated infusion pump that, based on previously defined blood glucose value that is previously defined, depending on the need, automatically administer the right amount of insulin or glucose”. Not always true: CGM devices are often used without insulin pumps.

Response 6: Thank the reviewer for this suggestion. We have specified that in this context CGM devices are used especially for glucose administration. Please, see the rows 504-506.

Point 7: Lines 479-481: “an integrated approach among these specialties to improve clinical management and outcomes is still missing.” I think that this integrated approach could be better implemented in some hospitals, but it is already working well in many others.

Response 7: Thank the reviewer for this comment. We have better specified our thoughts in the text.  “however, an integrated approach among these specialties to improve clinical management and outcomes, already working well in many hospitals, is still missing, could be better implemented in some others”. Please, see the rows 601-602.

Minor issues:

Point 1: Just from the abstract and throughout the manuscript, neuroendocrine tumors are abbreviated both as NETs and NENs. I suggest using just one acronym (eg NENs, more comprehensive and updated term)

Response 1: Thank the reviewer for this suggestion. We have revised accordingly from the abstract and throughout the manuscript.

Point 2: Abstract, line 21: “The former is mainly related to the serotonin secretion by a small bowel NET, in most cases located in the small bowel” (redundancy of “small bowel”)

Response 2: Thank the reviewer for pointing that out. We have eliminated the redundancy.

Point 3: Acronyms should be explained the first time they are used, and therefore should be used along the text:

    1. Introduction, Line 47: Carcinoid syndrome (to be replaced by CS)
    2. Introduction, Line 57: insulinoma (INS) could already be explained at Line 44, and therefore INS should be used throughout the manuscript

Response 3: Thank the reviewer, for pointing that out. We have revised accordingly.

Point 4: Discussion, line 492: “From a clinical point of view, the clinical benefit of the patient is strongly correlated…” (redundancy of “clinical”).

Response 4: Thank the reviewer, for pointing that out. We have revised accordingly. Please, see the row 613.

Point 5: I think that a recent review dedicated to CS deserves to be mentioned: Serotonin pathway in carcinoid syndrome: Clinical, diagnostic, prognostic and therapeutic implications. NIKE Group. Rev Endocr Metab Disord. 2020 Dec;21(4):599-612. doi: 10.1007/s11154-020-09547-8.

Response 5: Thank the reviewer, for pointing that out. We have added this reference. Please, see the row 51.

Round 2

Reviewer 1 Report

I appreciated the changes made in the paper but I have to underline that the diagnotic section needs to be updated expecially regarding the PET imaging role. The proposed indication for insulinoma have been overtaken by the actual practice and the more recent EANM guideline have de facto defined the 68Ga dotapeptide PET indication in insulinomas both for diagnosis and possible therapy. Moreover I remember that speaking of affinity profiles (IC50) for human sst1–sst5 receptors, toghether with the higher sensitivity, both the more used radiopharmaceutical demonstrated a crossed reactivity that is well known. 68Ga-dotatoc demonstrated an affinity for sst2 (2.5), for sst5 (73) and for sst3 (613), Ga-dotatate toghether with an higher sst2 sensitivity (0.2) has a relevant sst5 sensitivity of (377)*.

 In the same way, the morphologic imaging has improved his diagnostic tools with intraoperative ultrasonography and arterial stimulation venous sampling that needs to be illustrated.

*: All values are IC50±SEM in nM.

I would suggest some bibliographic note:

Jean Claude Reubi, Jean-Claude Schär, Beatrice Waser, Sandra Wenger, Axel Heppeler, Jörg S. Schmitt,

Helmut R. Mäcke. Affinity profiles for human somatostatin receptor subtypes SST1–SST5 of somatostatin radiotracers selected for scintigraphic and radiotherapeutic use. European Journal of Nuclear Medicine

Vol. 27, No. 3, March 2000

Irene Virgolini, Valentina Ambrosini, Jamshed B. Bomanji, Richard P. Baum, Stefano Fanti, Michael Gabriel, Nikolaos D. Papathanasiou, Giovanna Pepe, Wim Oyen, Clemens De Cristoforo, Arturo Chiti. Procedure guidelines for PET/CT tumour imaging with 68Ga-DOTA-conjugated peptides: 68Ga-DOTA-TOC, 68Ga-DOTA-NOC, 68Ga-DOTA-TATE. Eur J Nucl Med Mol Imaging (2010) 37:2004–2010 DOI 10.1007/s00259-010-1512-3

Diagnosis and management of insulinoma Takehiro Okabayashi, Yasuo Shima, Tatsuaki Sumiyoshi, Akihito Kozuki, Satoshi Ito, Yasuhiro Ogawa, Michiya Kobayashi, Kazuhiro Hanazaki. World J Gastroenterol 2013 February 14; 19(6): 829-837 ISSN 1007-9327 (print) ISSN 2219-2840 (online) © 2013 Baishideng.

Author Response

Thank the reviewer. Please, see the attachment.

Reviewer 2 Report

I carefully evaluated the changing in the article and I found them complete and appropriate. As far as I'm concerned no further change or modification is needed.

Author Response

Thank the reviewer for appreciating our efforts and for pointing out the deficiency in the diagnostic part. We revised the diagnostic section and integrated it accordingly. In particular:

  • inserted rows 206-208, 228-230, 232-239
  • eliminated rows 277-295
  • inserted rows 302-327

Please see the attachment (reviewed manuscript)

Round 3

Reviewer 1 Report

II consider the changes made in line with what has been requested